# Integration of Microfluidic Chip and Probe with a Dual Pump System for Measurement of Single Cells Transient Response

**DOI:** 10.3390/mi14061210

**Published:** 2023-06-07

**Authors:** Xu Du, Shingo Kaneko, Hisataka Maruyama, Hirotaka Sugiura, Masaru Tsujii, Nobuyuki Uozumi, Fumihito Arai

**Affiliations:** 1Department of Micro-Nano Mechanical Science and Engineering, Nagoya University, Nagoya 464-8603, Japan; du.xu.j1@s.mail.nagoya-u.ac.jp (X.D.);; 2Department of Mechanical Engineering, The University of Tokyo, Tokyo 113-8656, Japan; 3Department of Biomolecular Engineering, Graduate School of Engineering, Tohoku University, Sendai 980-8579, Japan

**Keywords:** microfluidic chip, liquid exchange, single cell, transient response, 3D-printed probe

## Abstract

The integration of liquid exchange and microfluidic chips plays a critical role in the biomedical and biophysical fields as it enables the control of the extracellular environment and allows for the simultaneous stimulation and detection of single cells. In this study, we present a novel approach for measuring the transient response of single cells using a system integrated with a microfluidic chip and a probe with a dual pump. The system was composed of a probe with a dual pump system, a microfluidic chip, optical tweezers, an external manipulator, an external piezo actuator, etc. Particularly, we incorporated the probe with the dual pump to allow for high-speed liquid change, and the localized flow control enabled a low disturbance contact force detection of single cells on the chip. Using this system, we measured the transient response of the cell swelling against the osmotic shock with a very fine time resolution. To demonstrate the concept, we first designed the double-barreled pipette, which was assembled with two piezo pumps to achieve a probe with the dual pump system, allowing for simultaneous liquid injection and suction. The microfluidic chip with on-chip probes was fabricated, and the integrated force sensor was calibrated. Second, we characterized the performance of the probe with the dual pump system, and the effect of the analysis position and area of the liquid exchange time was investigated. In addition, we optimized the applied injection voltage to achieve a complete concentration change, and the average liquid exchange time was achieved at approximately 3.33 ms. Finally, we demonstrated that the force sensor was only subjected to minor disturbances during the liquid exchange. This system was utilized to measure the deformation and the reactive force of *Synechocystis* sp. strain PCC 6803 in osmotic shock, with an average response time of approximately 16.33 ms. This system reveals the transient response of compressed single cells under millisecond osmotic shock which has the potential to characterize the accurate physiological function of ion channels.

## 1. Introduction

Robot-integrated microfluidic chips have immense potential for exploring the biophysical properties of cells. These chips integrate a series of functional components that allow for cell separation, capture, detection, and several other tasks for single cells [1,2,3,4,5,6,7,8,9,10,11,12,13]. For instance, an optical tweezers system was used to trap or stretch a single cell [2,6,8]. Additionally, on-chip functional probes were fabricated with the chips to puncture or deform the cell, providing insight into the mechanical properties of the cell [3,4,6,7].

The integration of liquid exchange is critical in biomedical and biophysical fields since it enables the switching of the extracellular environment, thereby exposing cells to stimulation and detection simultaneously [7,14,15,16,17,18,19,20,21,22,23,24,25]. This allows for the direct observation of detailed and dynamic cell responses. To achieve precise measurements of transient cell responses, it is necessary to have high-speed liquid exchange and accurate measurement.

To achieve liquid exchange in the microfluidic device, several approaches were examined in previous studies. Multiple laminar flows were commonly used in a closed chip to switch the solutions [7,14,15]. For instance, in our previous research, we investigated the response of mechanosensitive (MS) channels in *Synechocystis* sp. strain PCC6803 to osmolarity change [6,7]. MS channels are a type of ion channel that sense membrane tension force and release cytoplasm into the extracellular environment to prevent cell bursting in extracellular osmolarity change [26,27,28]. We designed a closed microfluidic chip that integrated liquid exchange and force measurement to investigate the dynamic response of *Synechocystis* under osmolarity change [7]. The liquid exchange time in our previous experiment was approximately 0.5 s, which was relatively slow compared to the reported response time of *Synechocystis* to osmolarity changes (approximately 20 ms) [27]. Therefore, the accurate response process of the cells was not fully captured in our previous work. A higher injection pressure was necessary to achieve much high-speed liquid exchange in the laminar flow method. Conversely, increased injection pressure caused much bigger disturbances to the force sensor of the microfluidic chip. There are a few other drawbacks that remain, and most closed microfluidic chips, sample processing, and analysis occurred within closed microchannels, which imposed restrictions on the use of external tools and sample preparation. In addition, it is also a challenge to control the position and pressure of liquid exchange within a closed microchannel.

Another approach is a probe with a pumps system. A probe with a pumps system has shown great promise in controlling fluid flow with a high response speed and high accuracy volume adjustment [16,17,18,19,20,21,29,30]. Recent research has demonstrated that this system was utilized to inject and withdraw solutions in an open space by simultaneously applying positive and negative pressure at adjacent barrels of a probe to form a hydrodynamically confined flow volume at the probe tip [16,18,19,20,21,22,23,24,25]. The advantage of this method is that the exchange solutions are achieved in a localized open space without disturbing the surrounding liquid [21]. Polydimethylsiloxane (PDMS) material-based probes were a common choice since they allow for the relatively easy fabrication of multi-barrel probes, and external pumps were used to control the flow volume [16,17,19]. A higher liquid exchange time was demonstrated, which was approximately 200 ms [17]. Another study achieved a millisecond response time using a probe with a single pump system (with no liquid exchange function) [30]. Hence, the liquid exchange time of 200 ms has not reached the full potential of this method. The problems might be attributed to the sophisticated built-in method, which elongated the distance between the outlet and pressure source, increasing the pressure drop in tubes. Furthermore, the soft material-based probe and tubes also extend the response time [30].

Therefore, we propose a liquid exchange system that is easily integrated with an open chip to enable high-speed, low-turbulence liquid exchange. The system consists of two piezo pumps and a 3D-printed probe compactly assembled to reduce the response time. Two barrels in the 3D-printed probe injected and sucked liquid, preventing diffusion before the 3D-printed probe connected to the original liquid. A 3D manipulator was utilized to position the probe tip near the upper surface of the open chip. Pushing and sensor probes were integrated into the open chip to measure cell deformation and reactive force. The liquid exchange time was evaluated using the time constant of grayscale level change which was approximately 3.33 ms. We measured the transient response of *Synechocystis* sp. strain PCC 6803, with an average response time of approximately 16.33 ms. Compared to previous methods, this system achieves millisecond liquid exchange with lower turbulence and represents a promising tool for investigating the properties of cells.

## 2. Materials and Methods

### 2.1. Overview of the On-Chip Cellular Measurement System

As shown in Figure 1a, the open chip was fixed on the microscope by a jig, and the cells were added to the surface of the chip using a pipette. The target cell on the surface of the open chip was trapped at the focal point of the laser (Yb fiber laser, IPG Photonics, wavelength: 1064 nm) and transported by controlling the position of the focal point with Galvano mirrors (GVS102, Thorlabs Japan Inc., Tokyo, Japan). The external piezo actuator was used to drive the pushing probe to compress the trapped single cell. To achieve the liquid exchange, we assembled a probe with a dual pump system, and a 3D manipulator (SMX, Sensapex, Oulu, Finland) moved the probe tip to a designated position close to the chip surface. A signal generator connected to an amplifier was used to excite and synchronize the two pumps. The displacement of on-chip probes and grayscale level changes were observed and recorded using a high-speed camera (VW-9000, KEYENCE CORPORATION, Osaka, Japan) with 1000 frames per second.

### 2.2. Probe with the Dual Pump System

The probe with the dual pump system consists of two piezo stacks, silicon tubes, glass tubes, and a 3D-printed probe with two adjacent barrels (Figure 1b). To achieve a high-speed response and liquid exchange, the length of the barrels and tubes should be minimized, while the stiffness of the tubes and probe should be as rigid as possible [20]. We utilized 3D printing technology to fabricate the probe with two adjacent barrels as it enables the creation of complex structures with multiple barrels. Additionally, the 3D-printed probe was compactly assembled with shorter silicon tubes and glass tubes to reduce the response time. Moreover, the resin material used in 3D printing was stiffer than PDMS. Owing to the size limitation of the high-precision 3D printer (BMF 130, BMF Precision Tech Inc., Shenzhen, China), the 3D-printed probe was divided into two parts for printing. The probe tip was fabricated by the high-precision 3D printer, and it could be inserted into the connector made by the other 3D printer (Foto 6.0, Zhejiang Flashforge 3D Technology Co., Ltd., Jinhua, China) and bonded with adhesive. Two piezo stacks were assembled with silicon and glass tubes to form pumps that provided each barrel of the 3D-printed probe with individual and picoliter-level volume control [21]. The driving signal of the two piezo pumps was synchronized using the signal generator. The displacement of piezo stacks deformed the silicon tubes, thus changing the volume of the solution in the silicon tubes. Hence, solutions were injected and sucked from the two holes in the probe tip synchronously, and the original solution between the holes of the probe tip was replaced with the injected solution. 

### 2.3. Robot-Integrated Microfluidic Chip

The fabricated microfluidic chip is shown in Figure 2a. Two on-chip probes were designed to capture and deform single cells (Figure 2b). To prevent leakage when adding the solution to the measurement area, the pushing probe was designed inside the chip without connecting to the external environment. Thus, the chip surface retained enough solution to decrease the concentration change caused by evaporation. The pushing probe was driven by an external piezo-actuator, and the reactive force of a single cell was simultaneously measured using a sensor probe composed of a hollow folded beam. Small holes were patterned on the on-chip probes which were used to generate moiré fringes that contributed to amplifying the measured displacement [4,6].

The fabrication process of the open chip was based on MEMS technology (Figure 2c). The open chip consisted of a cover glass and a silicon-on-insulator (SOI) wafer. An SOI wafer with a device layer of approximately 10 μm was selected to enable the measurement of *Synechocystis* cells (which have a diameter of approximately 2 μm). The handle layer of the SOI wafer had a thickness of approximately 400 μm, and the glass layer had a thickness of approximately 100 μm.

The fabrication process for the open chip is summarized as follows:(1)Spin-coating of SU-8 3010 (Nihon Kayaku Co. Ltd., Gumma, Japan) photoresist onto the glass surface, followed by patterning using a mask aligner.(2)Deep reactive ion etching (DRIE) of the glass to prevent friction between the on-chip probes and the glass.(3)Sputtering of a thin layer of Cr onto the etched side of the glass to protect the movable parts, followed by the removal of the photoresist and Cr using a piranha solution.(4)Spin-coating of the OFPR (Tokyo Ohka Co., Ltd., Tokyo, Japan) photoresist onto the device layer surface.(5)DRIE etching of the device layer, followed by the removal of the photoresist using a piranha solution.(6)Bonding of the glass with the device layer.(7)Spin-coating and patterning of SU-8 onto the handle layer surface.(8)DRIE etching of the handle layer, followed by the removal of SU-8 using oxygen plasma ashing.(9)Removal of the silicon dioxide layer using a buffered hydrogen fluoride solution, and cleaning of the chip with a piranha solution.(10)Removal of the Cr layers on the glass using chrome etchant.

### 2.4. Cell Culture and Preparation

All the *Synechocystis* cells used in this research were provided by Tohoku University. The cells were cultured in a BG11 medium at a temperature of 28 °C for 5 days under continuous illumination using 3 5 W white-light LED sources. BG11 solution was used as a low osmotic concentration (LOC) solution, and BG11 mixed with 0.5 mol·L^−1^ sorbitol was used as a high osmotic concentration (HOC) solution. Before the experiment, the cultured cells in BG11 were centrifuged at 620× *g* (where g is the gravitational acceleration) for 5 min and then resuspended in the HOC solution. Additionally, this process had the capability of decreasing the polysaccharide surrounding the cells, thereby reducing cell adhesion to the on-chip probes. To distinguish between the two solutions and make it possible to observe grayscale level changes during the liquid exchange, 10 g/L of Rhodamine B was mixed with the LOC solution.

### 2.5. Compression Concept, Force Sensor Calibration, and Stability of Force Sensor

Figure 3a shows the force measurement process using the on-chip probes. The target bead or cell was positioned between two on-chip probes by optical tweezers. The pushing probe was driven by an external piezo actuator, while the sensor probe was connected to a folded beam, functioning as a spring force sensor (Figure 3b,c). The start position of deformation is defined as the point where the sensor probe has a displacement; at this point, the distance between the on-chip probes is regarded as the original cell diameter *D*_0_. Then, the pushing probe was gradually pushed to deform the target. The reactive force is calculated based on the displacement of the sensor probe *δ_s_* and calibrated spring constant *k* of the force sensor. The deformation of the target is calculated by measuring the difference in displacement between the two on-chip probes.

The spring constant of the force sensor beam was calibrated using PDMS beads (Figure 3d). The specific fabrication process of the beads was previously reported in our work [6].

The deformation and reactive force of the beads were measured using a microfluidic chip to calibrate the spring constant of the force sensor. 

The Hertzian model can be used to describe the relationship between the deformation of a sphere and the force of non-adhesive elastic contact [31,32]. The equation can be simplified as follows:(1)$$\;\;F=\frac{4{\left(D_{0}/2\right)}^{1/2}}{3}\cdot \frac{E_{p}}{1-v^{2}}\cdot {\left(\frac{\delta_{c}}{2}\right)}^{\frac{3}{2}}\;\;\;$$
where *F* represents the reactive force (N); *k* represents the spring constant of the force sensor (N·m^−1^); *E_p_* is Young’s modulus of the beads (Pa); and *ν* is Poisson’s ratio of the cell. Poisson’s ratio is taken to be 0.5 by assuming that the cell is composed of incompressible materials. The deformation *δ_c_* and the reactive force can be calculated using Equations (2) and (3):(2)$$\;\;\;\;F=k\delta_{s}\;\;\;\;$$
(3)$$\;\;\;\delta_{c}=\delta_{p}-\delta_{s}\;\;$$

The spring constant of an end-loaded thin beam with a rectangular cross-section can be estimated using the following equation:(4)$$\;\;k=\frac{w^{3}hE_{b}}{L^{3}}\;\;\;\;\;$$
where *E_b_* is Young’s modulus of the beam (Pa) and *L*, *w*, and *h* represent the length, width, and thickness of the rectangular beam, respectively (m). In this study, Equation (4) is used to evaluate the spring constant of the beam.

In Figure 3d, the deformation ratio (%) in the x-axis represents the ratio of the deformation of the bead to its diameter. The orange curve shows the theoretical values using Equation (1). The blue dots represent the deformation of the PDMS beads measured using the microfluidic chip, and the corresponding reactive force was calculated using Equation (2), and the *k* was adjusted to fit the theoretical curves. The calibrated spring constant *k* was 0.07 N m^−1^, and the standard deviation (SD) was ±0.01 N m^−1^.

We calculated the reactive force by measuring the displacement of the sensor probe. As such, the resolution of the force measurement was contingent upon the resolution of the displacement measurement. To assess the accuracy of the force sensor, we recorded the measurement noise for the displacement of the sensor probe in the experimental environment for a duration of 0.5 s (Figure 3e). The stability of the force sensor was evaluated using 3 times the standard deviation (*σ*), which was 42.45 nm, corresponding to a force resolution of 3.10 nN.

## 3. Results and Discussion

### 3.1. Liquid Exchange Process

The liquid exchange time is dependent on the response time of the probe with the dual pump system and the diffusion extent of the injected solution before the complete replacement of the extracellular environment. As we discussed above, to achieve a high response time, a probe with a dual pump system was compactly assembled, and the length of the barrels and tubes was minimized, while the stiffness of the tubes and probe was maximized [30]. Before starting the experiment, it is essential to pre-position the relative position of the on-chip probes and probe tip to reduce the time needed for the probe tip position during the experiment. This reduces the contact time between the injected solution in the probe tip and the original solution on the surface of the open chip, thereby reducing diffusion between the two solutions. 

The pre-positioning process is described as follows. First, we used the jig to fix the chip on the microscope. Then, we moved the microscope’s two-dimensional movement platform until the on-chip probes (measurement areas) were within the camera’s observation field and marked their positions. Next, we removed the chip and placed a glass plate with a thickness similar to the chip to prevent damage to the on-chip probes. After that, a 3D manipulator was used to control the probe tip to reach the midpoint of the marked positions of the two on-chip probes. We recorded the current position using a 3D manipulator (positioned in the horizontal direction). Next, we replaced the glass plate with the chip and fixed the on-chip probes and probe tip in the horizontal direction, adjusting the distance between the top surface of the on-chip probes and the probe tip to approximately 6 μm (positioned in the vertical direction). Finally, we recorded the current position of the probe tip and then pulled the probe tip away from the surface of the chip along the vertical direction.

After the pre-positioning process, the LOC solution was injected into the injection barrels, and the HOC solution was injected into the suction barrels. Both solutions were injected from the connector of the probe with the dual pump system to remove all the bubbles in the tubes and barrels. The HOC solution with cells was added to the surface of the open chip. As shown in Figure 4, the optical tweezers system was used to trap a single cell and position it in the middle of the two on-chip probes. At this point, the cell was immersed in the HOC solution. Subsequently, we excited the external piezo actuator to push the pushing probe to compress the single cell. Once the compression of the cell commenced, we drove the 3D manipulator to move the probe tip to the pre-positioned location. Then, the ascending trapezoidal wave was applied to the suction pump, causing the piezoelectric stack to extend and inject the HOC solution around the cell. This step aimed to maintain a constant concentration of the HOC solution around the cell and keep the piezo stack in an extended state. After approximately 1 s, we applied an ascending trapezoidal wave to the injection pump while synchronously adjusting the applied voltage of the other pump to zero. The LOC solution was injected from the injection barrel, and the liquid below the probe tip was simultaneously aspirated from the other barrel. Owing to the large volume of injected liquid compared to the volume of liquid below the probe tip, the extracellular environment was replaced from HOC solution to LOC solution. The synchronization of the injection and suction was verified in our previous work [33]. During the liquid exchange process, the trapped cell swelled and the pushing probe remained stationary, allowing the displacement of the sensor probe to be used to calculate the dynamic deformation and reactive force of the single cell. Moreover, Rhodamine B was added to the LOC solution to alter the brightness to detect liquid exchange using gray scale level change. After a few seconds, the pushing probe was released by the external piezo actuator, and the probe tip was moved away from the surface of the chip along the vertical direction.

To evaluate the effect of the position and area of analysis region on the liquid exchange time, we analyzed changes in the gray scale level at different locations during the liquid exchange process, as shown in Figure 5a. To achieve the liquid exchange, the applied injection and suction voltages were altered from 0 to 30 v and 10 to 0 v, respectively (in the subsequent sections, the injection voltage changes from 0 to *a* are abbreviated as the injection voltage *a*, and likewise, the suction voltage changes from *b* to 0 are abbreviated as the suction voltage *b*), with a rising time of 1 ms. To eliminate the influence of background brightness on the analysis of the trend of grayscale value changes, we normalized the grayscale value changes on the y-axis in Figure 5b. The time constant *τ* represents the time for one system’s step response to reach approximately 63.2%, and it is used to characterize the liquid exchange time or the response time of the cell [27]. The fitting curve of the gray scale level data indicated the changes in the liquid exchange time at different positions [34]. At positions far away from the suction hole (indicated by the purple circle), the grayscale value remained almost unchanged. The liquid exchange time became shorter as the analysis region got closer to the injection hole. The yellow and green circles were located near the trapped cell, and a longer liquid exchange time was observed at the green circle, and the obstruction of fluid flow by on-chip probes is a possible reason. The liquid exchange time was shorter than 10 ms in all the selected analysis regions which demonstrated the achievement of millisecond liquid exchange by using this system. To ensure the reliability of the results, we estimated the liquid exchange time at the green circle position in subsequent experiments. Additionally, we further investigated the effect of the area of analysis region on grayscale value changes (Figure 5c). At the green circular position, we evaluate the grayscale value changes during the liquid exchange in concentric circular areas with diameters of 2, 4, and 6 μm. To illustrate the influence of background brightness on the analysis results, changes in grayscale values were not normalized in this figure. Owing to the background brightness changes at different positions, the curve had a slight shift, but the obtained liquid exchange times showed no obvious change. Therefore, the area of analysis position does not significantly affect the analysis result of the liquid exchange time.

Based on the above analysis, we further investigated the effects of injection voltages on the liquid exchange degree (Figure 6a). In this experiment, the suction voltage was 10 v, and other parameters were consistent with the experimental conditions described above. The applied voltage and the corresponding injected liquid volume are shown in the x-axis of Figure 6a. The injected liquid volume corresponding to the injection voltage was calculated based on the change in the volume inside the silicon tube caused by the extension of the piezo stack. When the injection voltage was less than 20 v, the change in the grayscale value before and after the liquid exchange increased as the injection voltage increased. This indicated that the injected liquid volume was not sufficient to replace the diffused solution inside the injection barrel and the original solution surrounding the cell. When the injection voltage exceeded 20 v, the change in the grayscale value tended to be stable, which means the extracellular solution was completely exchanged. The injected liquid volume required for complete liquid exchange is dependent on the degree of diffusion which is correlated to the duration of contact between the injected solution (LOC) and the original solution on the surface of the chip (HOC). Therefore, the experiment should minimize contact time between the two solutions. In this experiment, the experimental procedure was optimized, and the contact time between the two solutions was reduced to approximately 3 s. In addition, it is also essential for the injected liquid volume to be significantly greater than the volume of liquid surrounding the cell. The liquid volume between the holes of the probe tip and the chip bottom surface was approximately 100 pL which was much smaller than that of the injected liquid volume. Given that the duration cannot be accurately controlled, we chose a larger injection voltage of 30 v to ensure complete liquid exchange in the following experiment.

To evaluate the disturbance of the sensor probe under liquid exchange accurately, we replaced the cells with PDMS beads that have a similar Young’s modulus and size as *Synechocystis* cells since the volume of PDMS beads is not influenced by osmolarity change. The applied injection voltage was 30 v, and the suction voltage was 10 v in the liquid exchange. The liquid exchange occurred from approximately 0.097 s to 0.102 s. The disturbance of the liquid exchange was almost impossible to observe in the displacement data of the sensor probe (Figure 6b). Compared to the disturbance of the liquid exchange in a closed chip using the laminar flow method [7], the probe with the dual pump system achieved localized liquid exchange, which significantly decreases the disturbance under the liquid exchange process and increases the measurement accuracy of the force sensor.

### 3.2. Measurement of Synechocystis Cells

We measured the transient response of single cells using a system integrated with an open chip and a probe with a dual pump. The captured cells were compressed by on-chip probes, resulting in a deformation of approximately 20–30%. The probe tip was then positioned 6 μm above the surface of the on-chip probes. The injection voltage was 30 v while the synchronized suction voltage was 10 v to achieve the high-speed liquid exchange. Thus, the extracellular environment of the cell was replaced from the HOC solution to the LOC solution mixed with Rhodamine B. The high-speed camera recorded the displacement of the on-chip probe and simultaneously recorded the changes in brightness within the measurement area. Figure 7 shows one example of the transient response of a single cell and the corresponding gray scale level change in the liquid exchange process. We ignored the displacement of the pushing probe since it was stopped when the liquid exchange happened, and the end of the force sensor was fixed. Hence, the deformation of the single cell was the same as the displacement of the force sensor in the liquid exchange process. The cell size ratio in the vertical axis label of Figure 7a represents the ratio of the cell diameter after deformation to the original diameter. The cell responded to osmolarity change and swelled. The time constant of the response of this single cell was approximately 19 ms. Meanwhile, the liquid exchange time was evaluated using the gray scale level change time (Figure 7b), and the time constant was approximately 4 ms. We measured 9 cells, and the average value of the transient response times of the cells was 16.33 ms, while the standard deviation was ±4.47 ms. Additionally, the corresponding average value of the liquid exchange times was 3.33 ms, and the standard deviation was ±1.50 ms. The liquid exchange time is much faster than that of the response time of cells. 

The average liquid exchange time of the system is over 100 times faster than our previous work and also faster than the response time of cells to osmolarity change [7]. Hence, we successfully exposed the process of a single compressed cell responding to osmolarity change by using a proposed system. Furthermore, compared to the reported response time of a large number of *Synechocystis* cell volume changes [27], the measured response time in this research is reasonable, and we further achieved accurate contact measurements of the force and deformation measurements of a compressed single cell.

## 4. Conclusions

In this study, we developed a system integrated with a microfluidic chip and a probe with a dual pump to investigate the transient response of single *Synechocystis* cells. The proposed system allows for a fast and localized liquid exchange on the chip surface with minimal disturbance. The average liquid exchange time of the system is approximately 3.33 ms, which is over 100 times faster than our previous work and also faster than the response time of cells to osmolarity change, which enables the accurate exposure of the transient response of single cells. 

Given that MS channels sense the membrane tension force, the compression or expansion of cells can reveal the characterization of MS channels. In the future, we will conduct a comparison between the response of wild-type and mutant-type (MS channel defect) *Synechocystis* cells under high-speed osmolarity change to expose the physiological function of MS channels. Furthermore, excessive compression force may damage the cells. In future experiments, we will estimate the impact of compression force on cells. Overall, our results demonstrate the effectiveness of the developed system for studying single cell responses accurately and provide insights for future research in the biophysical and biomedical fields.

## Figures and Tables

**Figure 1 micromachines-14-01210-f001:**
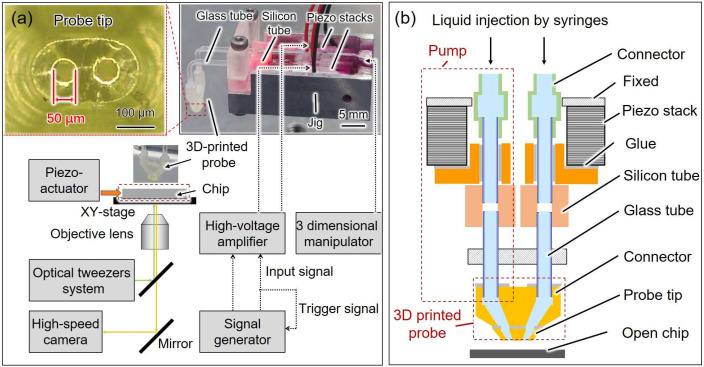
(**a**) Experimental setup and the image of the probe with the dual pump system and probe tip; (**b**) schematic of the proposed probe with the dual pump system.

**Figure 2 micromachines-14-01210-f002:**
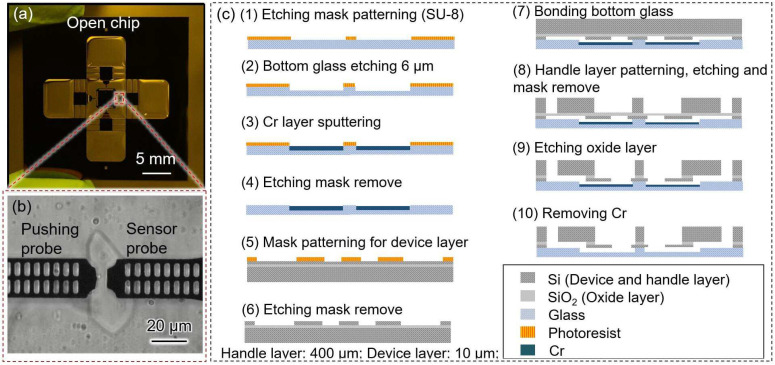
The image and fabrication process of the robot-integrated microfluidic chip. (**a**) The image of the fabricated open chip; (**b**) the microscope image of the measurement area; (**c**) the fabrication process of the open chip.

**Figure 3 micromachines-14-01210-f003:**
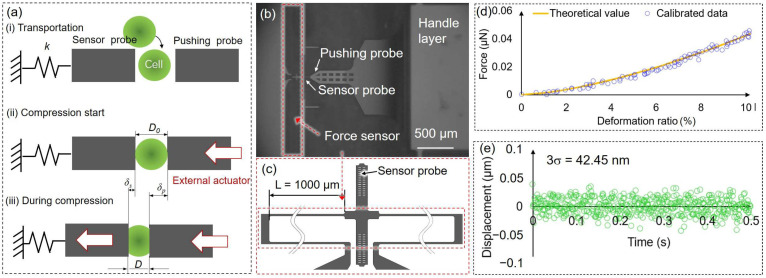
(**a**) Concept of the force measurement process; (**b**) image of the pushing probe and the force sensor; (**c**) the hollow folded beam structure functioning as a spring force sensor; (**d**) an example of the calibration data and theoretical value of the force sensor (the blue dots show the measurement date using the force sensor after calibration, and the orange curve represents the theoretical values); (**e**) the stability of the force sensor was evaluated by measuring the 3*σ* of displacement data of the sensor probe. *D*_0_: the original cell diameter; *δ_s_*, *δ_p_*: the displacement of the sensor and pushing probe; *L*: the length of the beam structure; *σ*: standard deviation.

**Figure 4 micromachines-14-01210-f004:**
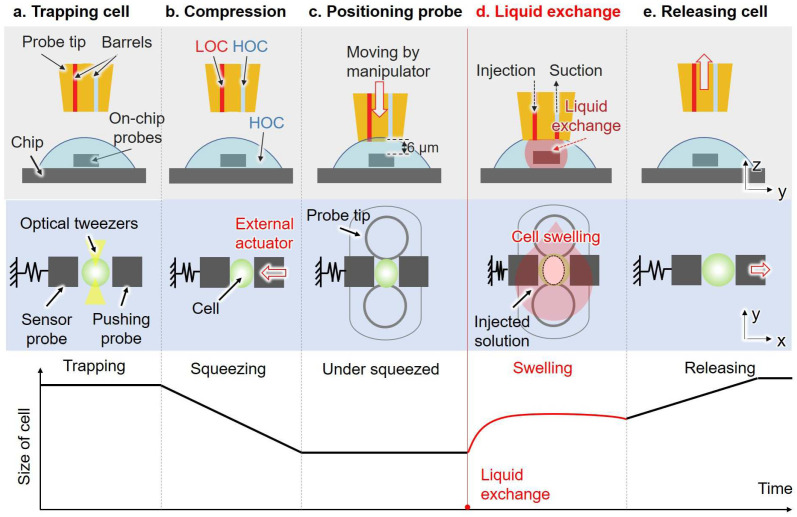
The liquid exchange process with a curve underneath showing the schematic of cell size change corresponding to each step. (**a**) The optical tweezers system is used to trap a single cell and position it in the middle of the two on-chip probes; (**b**) the target cell is compressed using an external piezo actuator; (**c**) the probe tip is moved to the pre-positioned location using a 3D manipulator; (**d**) the LOC solution is injected from the injection barrel, and the liquid below the probe tip is simultaneously aspirated from the suction barrel, while the extracellular environment changes from the HOC solution to the LOC solution; (**e**) the pushing probe is released by the external piezo actuator, and the probe tip is moved away from the surface of the chip along the vertical direction.

**Figure 5 micromachines-14-01210-f005:**
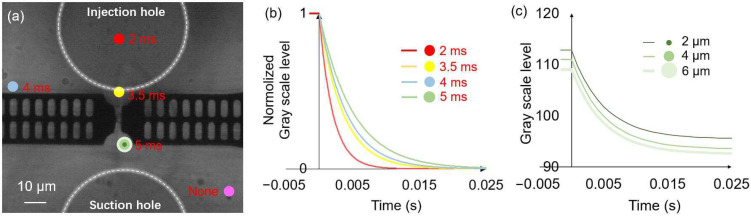
The effect of the position and area of analysis region on the evaluation of the liquid exchange time. (**a**) Microscope image of the measurement area. The circles in different colors indicate various positions for analyzing the gray scale levels, and the corresponding liquid exchange times are described adjacent to the circles. (**b**) The effect of the position of the analysis region on the evaluation of liquid exchange time. The color of the fitting curve corresponds to that of the analysis region. (**c**) The analysis of grayscale value changes during the liquid exchange in concentric circular areas (green circles) with diameters of 2, 4, and 6 μm.

**Figure 6 micromachines-14-01210-f006:**
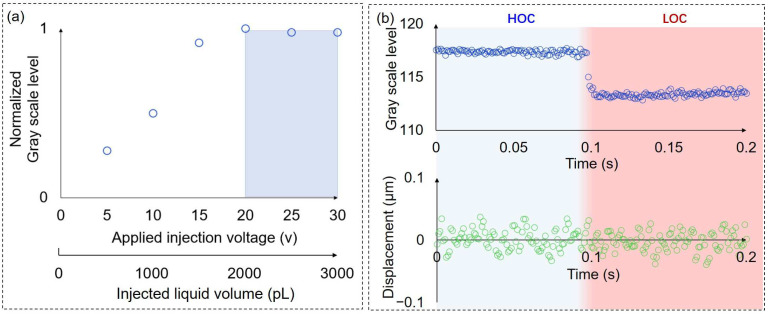
The effects of injection voltages on the liquid exchange degree and the disturbance of liquid exchange to the displacement measurement. (**a**) The effects of injection voltages on the liquid exchange degree. When the injection voltage exceeds 20 v, the change in the grayscale value tends to be stable, which means the extracellular solution is completely exchanged; (**b**) the gray scale level change and the corresponding disturbance of the sensor probe under liquid exchange.

**Figure 7 micromachines-14-01210-f007:**
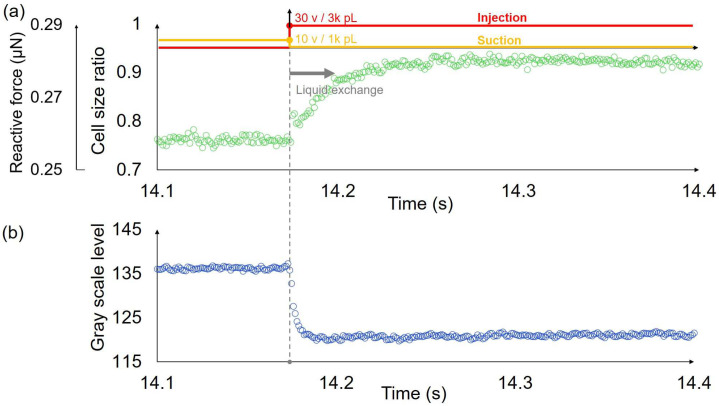
One example of the transient deformation of a single cell and the corresponding gray scale level change in the liquid exchange process. (**a**) The deformation and corresponding reactive force of a single cell in high-speed osmolarity change. The time constant is approximately 19 ms. (**b**) The gray scale level change of the location close to the trapped single cell in the liquid exchange process. The time constant is approximately 4 ms.

## Data Availability

Data is contained within the article.

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
