# Peer review of "Integration of Microfluidic Chip and Probe with a Dual Pump System for Measurement of Single Cells Transient Response"

_micromachines, 2023, doi:10.3390/mi14061210_

Round 1
Reviewer 1 Report
The submitted manuscript reports the evaluation of microfluidic chip and probe with a dual pump for single-cell analysis. Although the manuscript is interesting in the research field of technologies for single-cell analysis, the interpretation is seem to be not enough for explanation about advantage and disadvantage of the microfluidic system. Therefore the manuscript can be acceptable after consideration of following points.
Specific points of criticism:
1. Although the microfluidic system has a pushing probe for single-cell trapping, how is the damage and effect on trapped single-cell? The reviewer suggests the authors to describe and explain the effect of attach or stimulation of the pushing-probe on single cell analysis.
2. Since the microfluidic system was composed of a probe with the dual pump system, a microfluidic chip, optical tweezers, an external manipulator and so on, it will become complicated and difficult-to-use system for practical use in the future. How could you improve the microfluidic system for practical real application?
Author Response
Dear Reviewer
Thank you very much for reviewing my paper with interest. We have undertaken the works which reviewers required and suggested kindly. Here we revised our manuscript based on the reviewer’s advice. Please see the attachment.
Sincerely yours,
All authors

Reviewer 2 Report
This paper proposes a refined microfluidic design that can measure the transient response of a single cell, which is a very interesting work.
1. In Line 181, “mol·L−1” is a clerical error.
2. In Figure 3d, what does “deformation ratio” mean? How the theoretical and calibration values were obtained requires a clearer explanation.
3. In Line 214, the values of “mean” and “standard deviation” should be the same number of decimal places.
4. Will liquid flow during liquid exchange affect sensing detection?
Author Response

(The authors gave the same response as above.)

Reviewer 3 Report
In this manuscript, the author presents a novel system for measuring the transient response of single cell. The system consists of a probe for force sense, a probe for pushing cell and a double-barreled pipette for exchange liquid. The applied injection voltage for liquid exchange is optimized, which is approximately 3.3 ms. Finally, the author demonstrates the measure of the deformation and the reactive force of Synechocystis sp. strain PCC 6803 in osmotic shock. The manuscript is well written and the experiments is well organized, showing that the authors have well considered the subject. I think this manuscript could be accepted for publication in Micromachines.
Author Response
Dear Reviewer
Thank you very much for reviewing my paper with interest. We highly appreciate your support and it motivates us to continue our efforts. Thank you once again for your valuable support!
Sincerely yours,
All authors